# Association of cancer with overactive bladder and impact of overactive bladder on mortality among cancer survivors: NHANES 1999-2018

**Bin Zhi[1]☯‡, Pin Zhao[2]☯‡, Liyuan Duan[2], Yuan Liu[2], Zhaowei Zhu◉[2]\*, Xuepei Zhang[2]\***

**1** Department of Pharmacy, The First Affiliated Hospital of Zhengzhou University, Zhengzhou, Henan, PR China, **2** Department of Urology, The First Affiliated Hospital of Zhengzhou University, Zhengzhou, Henan, PR China

☯ These authors contributed equally to this work.
‡ These authors are joint senior authors on this work.
\* zzwdoctor6@126.com (ZZ); zhangxuepei@263.net (XZ)

⊕ OPEN ACCESS

## Abstract

### Background

Cancer is one of the leading causes of death worldwide. There is little knowledge on the association between cancer and risk of overactive bladder (OAB). Additionally, the impact of OAB on mortality among cancer survivors remains unknown. We aimed to examine the association between cancer and risk of OAB and investigate OAB associations with mortality outcomes in cancer survivors.

### Methods

We analyzed data of the National Health and Nutrition Examination Survey from 1999 to 2018. Cancer status was identified through the NHANES questionnaire. OAB was diagnosed with a cumulative OAB Symptom Score ≥ 3. Multivariable logistic analysis was performed to examine the relationship between cancer and OAB. Cox regression analysis was carried out to investigate the relationship between OAB and mortality in subjects with cancer.

### Results

In total, this study included a cohort of 32,166 participants. Within the study population, the occurrence of cancer was observed in 9.606%, and OAB was prevalent in 20.167%. The findings from the logistic regression analyses indicated a heightened risk of OAB among individuals with cancer in comparison to those devoid of cancer. Results from subgroup analyses also revealed affirmative correlations between cancer and OAB. Furthermore, the diagnosis of cancer, irrespective of whether it was genitourinary, non-genitourinary, pelvic, or non-pelvic in nature, was found to be correlated with an increased risk of OAB. Within the cohort comprising 3,090 participants diagnosed with cancer, a total of 850 all-cause deaths were recorded during a median follow-up duration of 76

**Data availability statement:** All relevant data are within the manuscript and its Supporting information files.

**Funding:** The author(s) received no specific funding for this work.

**Competing interests:** The authors have declared that no competing interests exist.

months. After accounting for multiple covariates, OAB was linked to an elevated risk of all-cause and cancer-specific mortality. Moreover, the influence of OAB on mortality from all causes was observed across various cancer types, including both genitourinary and non-genitourinary, as well as pelvic and non-pelvic cancers.

## Conclusions

The investigation identified a noteworthy positive correlation between cancer and the likelihood of OAB. Moreover, OAB was significantly correlated with an elevated risk of mortality among individuals who had received a cancer diagnosis.

## Introduction

Cancer is one of the leading causes of death worldwide and imposes an enormous economic burden on public health systems [1]. From 2000 to 2017, the cancer incidence increased for all cancers combined among young adults (ages 20-49 years) in China [2]. In 2024, 2,001,140 new cancer cases and 611,720 cancer deaths are projected to occur in the United States [3]. Cancer survival rates have substantially increased over the past few decades due to innovative surgeries, chemotherapy, targeted therapy, immune therapy and radiotherapy. The number of cancer survivors worldwide has been rapidly increasing in recent years, and the global cancer burden is expected to be 28.4 million cases in 2040, a 47% rise from 2020 [1]. Although modern medicine has greatly increased the lifespan of cancer survivors, there remains a persistent challenge in improving their long-term outcomes.

Overactive bladder (OAB) is defined as a syndrome characterized by storage symptoms by the International Continence Society in 2002. The cardinal features encompass urgency, with or without urge urinary incontinence (UUI), and are frequently associated with an elevated frequency of daytime voiding and occurrences of nocturia [4]. It has been reported that the overall prevalence of OAB stands at 11.8%. Furthermore, the prevalence rates for OAB characterized as ≥ "sometimes" and ≥ "often" were discerned as 17% and 8% in men, and 30% and 20% in women, respectively [5].

Cancer survivors are at a higher risk of developing OAB due to older age, surgeries, radiotherapy and various drugs. However, the prevalence of OAB among cancer survivors remains unknown. Additionally, few studies have focused on the relationship between OAB and mortality risk among cancer survivors. Therefore, there is a pressing need to identify modifiable risk factors among cancer patients, raise awareness about OAB in this population, and provide effective treatment for these distressing symptoms. By addressing these issues, we can enhance the quality of life for cancer survivors, boost their survival rates, and alleviate the associated socioeconomic burden. In light of these considerations, we conducted this cohort study to assess the association between cancer and OAB and to delve into the potential ramifications of OAB on mortality risk among cancer survivors.

## Methods

### Study design and participants

NHANES, a cross-sectional program crafted to assess the overall health and nutritional status of the US population, employs a blend of interviews and physical examinations. This methodology ensures the acquisition of comprehensive data encompassing demographics, dietary habits, medical examinations, laboratory results, and questionnaire responses. In the present study, we leveraged expansive datasets encompassing NHANES cycles spanning

from 1999-2000 to 2017-2018. The analytic population included adults who had complete information on cancer status, OAB, socioeconomic and health-related characteristics. During the participant selection process, we encountered a notably high percentage of missing data, which may exceed 40%. This is largely attributable to the inherent limitations of the NHANES database.

## Cancer status

The NHANES dataset encompasses a comprehensive classification of cancers, numbering up to 30 types. Cancer survivors were identified through the NHANES questionnaire: "Has you ever been told by a doctor or other health professional that you had cancer or a malignancy of any kind?" Those who responded "Yes" or "No" to the question were included in the study. The reliability and accuracy of self-reported cancer diagnoses in NHANES have been assessed in prior studies [6,7].

## Outcome variable

OAB has been characterized as an aberrant micturition reflex, distinguished by manifestations such as UUI and nocturia. All data presented herein emanates from questionnaires administered by proficient research personnel, conducted through methodical face-to-face interviews. The evaluation of urge incontinence severity involved the administration of two specified questions:

1. "During the past 12 months, have you leaked or lost control of even a small amount of urine with an urge or pressure to urinate and you could not get to the toilet fast enough?" [8]

2. "How frequently does this occur?" [8]

Additionally, the assessment of the severity of nocturia was conducted using a specific inquiry "During the past 30 days, how many times per night did you most typically get up to urinate, from the time you went to bed at night until the time you got up in the morning" [9]. We systematically assessed OAB symptoms utilizing the Overactive Bladder Symptom Score (OABSS), a validated questionnaire by Blaivas and colleagues [10]. Subsequently, the OABSS for each participant in the NHANES was computed by summing the UUI score and the nocturia score. The subjects with a cumulative OABSS ≥ 3 were categorically diagnosed with OAB.

## Mortality ascertainment

The National Center for Health Statistics database provides access to the NHANES public-use linked mortality files, encompassing data up to December 31, 2019. These mortality files have been cross-referenced with the National Death Index, enhancing the integrity and comprehensiveness of mortality-related research endeavors. To confirm survival status, a comprehensive set of 12 characteristics was employed to establish connections between NHANES samples and the National Death Index. For participants who survived, the survival time (in months) was computed by delineating the interval between the date of interview initiation and either the occurrence of mortality or the culmination of the follow-up, which concluded on December 31, 2019.

## Covariate assessment

In this study, control variables were identified to mitigate potential confounding influences. Among these, socioeconomic and health-related factors were particularly

scrutinized. Within the realm of socioeconomic determinants, variables such as sex and age were examined for their potential impact on the study's outcomes. Health-related features encompassed body mass index (BMI), smoking behaviors, and levels of alcohol intake. Categorizing these health-related features into distinct groups aligns well with established clinical guidelines and enhances the interpretability of results for both researchers and practitioners.

Within the scope of this investigation, hypertension was characterized by the presence of sustained elevations in systolic blood pressure (BP) meeting or surpassing 140 mmHg, concomitant with a diastolic BP averaging 90 mmHg or greater, or being under antihypertensive medication. The criteria for defining blood pressure values in hypertensive individuals are consistent with those used in numerous previous studies, treatment guidelines, and clinical practice. Diabetes was characterized by having a diagnosed condition of diabetes, current use of glucose-lowering medication, or meeting criteria such as HbA1c $\geq$ 6.5%, fasting serum glucose level $\geq$ 7.0 mmol/L, or random blood glucose $\geq$ 11.1 mmol/L [11].

## Statistical analysis

For NHANES datasets, the use of sampling weights and sample design variables is recommended for all analyses because the sample design is both a clustered design and incorporates differential probabilities of selection (https://wwwn.cdc.gov/nchs/nhanes/tutorials/samplede-sign.aspx). As highlighted on the CDC/NHANES website, in light of NHANES' intricate, multistage, stratified, cluster sampling design, the data analysis process employed suitable weighting procedures to derive nationally representative estimates [12]. The foundational demographic and clinical profiles were synthesized and organized into distinct strata, delineated by the presence or absence of OAB. Weighted means ± standard errors were employed for continuous variables, while unweighted counts with weighted percentages were used to present categorical variables. Comparative assessments across distinct cohorts necessitated the employment of statistical methodologies tailored to the nature of the variables under scrutiny. Specifically, weighted t-tests were employed for continuous variables, while chi-square tests were utilized for categorical variables.

The investigation employed multivariate logistic regression models to derive odds ratios (ORs) alongside their associated 95% confidence intervals (CIs), thereby evaluating the relationship between cancer status and OAB while accounting for sociodemographic covariates. Model 1 reflected the unadjusted crude model, and Model 2 entailed adjustments for sex, age, race, education level, marital status, and BMI category. Model 3 expanded upon these adjustments by integrating additional covariates including smoking status, drinking status, diabetes, and hypertension, thereby enhancing the depth and specificity of the analysis. We categorized cancer patients into two distinct groups: those afflicted with genitourinary (GU) cancer and those with non-GU cancer. Furthermore, the study participants were categorized into two groups based on the location of their cancer: those with pelvic cancer and those with non-pelvic cancer.

In this investigation, the statistical framework of multivariate Cox proportional hazards regression models was harnessed to quantify hazard ratios (HRs) alongside their respective 95% CIs. These models were instrumental in scrutinizing the relationship between OAB and mortality among cancer survivors across three distinct models. Model 1 represented the unadjusted crude model, and Model 2 involved adjustments for sex, age, race, education level, marital status, and BMI category. Model 3, characterized by a heightened level of comprehensiveness, extended its adjustments to encompass additional covariates such as smoking status, drinking status, hypertension, and diabetes.

The statistical analyses were conducted using the R software platform (version 4.2.2). The criterion for statistical significance was met when observing a two-sided P value below 0.05.

## Results

### Population characteristics between groups

The inclusion and exclusion criteria of the current study are outlined in Fig 1. As illustrated in Fig 1, this study included a cohort of 32,166 participants. Within the study population, the occurrence of cancer was observed in 9.6%, and OAB was prevalent in 20.2%. Notably, among those with OAB, the majority had an OAB score of 3, with fewer patients having scores of 4 or higher. Although the OAB values do not conform to a normal distribution, this pattern is highly representative of the clinical reality. Table 1 presents a thorough depiction of the characteristics of these participants, categorized by OAB status. Notably, among the 3,090 individuals diagnosed with cancer, 1,044 had a historical record of OAB. Subjects with OAB exhibited a mean age of 58.2 years, which was higher than those without OAB (45.4 years). Furthermore, individuals with OAB demonstrated a higher BMI in comparison to their counterparts lacking OAB (31.2 vs. 28.7, $P < 0.0001$). The prevalence of OAB was observed to be greater in participants with cancer and in those with a history of smoking and drink consumption, as well as among patients with diabetes and hypertension. The findings suggest that individuals with cancer constitute a more vulnerable demographic, exhibiting a higher prevalence of morbidity and a greater number of risk factors associated with OAB. It is crucial to be vigilant for signs of OAB, especially in patients where its occurrence is already suspected.

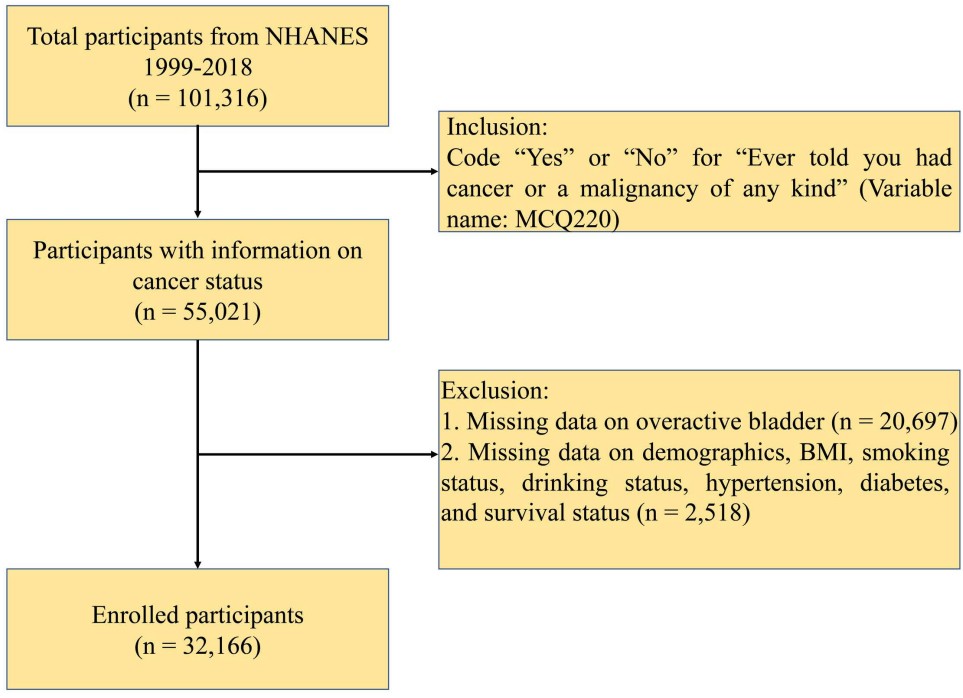

**Fig 1. Flowchart of the study population.**

**Table 1. General characteristics of individuals included in the NHANES survey stratified by the presence or absence of overactive bladder.**

| Characteristics | Total (n = 32,166) [a] | Individuals without overactive bladder (n = 25,679) [a] | Individuals with overactive bladder (n = 6,487) [a] | P value |
|---|---|---|---|---|
| BMI, mean ± SE | 29.06 ± 0.08 | 28.66 ± 0.09 | 31.17 ± 0.15 | < 0.0001 |
| Cancer | | | | < 0.0001 |
| No | 29076 (89.82) | 23633 (91.31) | 5443 (81.80) | |
| Yes | 3090 (10.18) | 2046 (8.69) | 1044 (18.20) | |
| Age group no. (Weighted%) | | | | < 0.0001 |
| <=49 | 16096 (55.17) | 14540 (60.26) | 1556 (27.86) | |
| 50-65 | 8433 (26.64) | 6342 (25.37) | 2091 (33.51) | |
| >=65 | 7637 (18.18) | 4797 (14.38) | 2840 (38.63) | |
| Race no. (Weighted%) | | | | < 0.0001 |
| Hispanic | 3074 (5.38) | 2433 (5.38) | 641 (5.38) | |
| Non-Hispanic White | 13850 (68.26) | 11235 (68.87) | 2615 (65.01) | |
| Non-Hispanic Black | 6882 (10.92) | 5019 (9.81) | 1863 (16.85) | |
| Mexican American | 5007 (8.32) | 4060 (8.49) | 947 (7.37) | |
| Other | 3353 (7.12) | 2932 (7.44) | 421 (5.39) | |
| Education no. (Weighted%) | | | | < 0.0001 |
| Less than high school | 7805 (15.50) | 5543 (13.75) | 2262 (24.92) | |
| High school or equivalent | 7401 (23.23) | 5871 (22.81) | 1530 (25.49) | |
| Some college or AA degree | 9547 (31.67) | 7789 (31.90) | 1758 (30.44) | |
| College graduate or above | 7413 (29.59) | 6476 (31.54) | 937 (19.15) | |
| Marital status no. (Weighted%) | | | | < 0.0001 |
| Divorced | 3559 (10.43) | 2616 (9.82) | 943 (13.70) | |
| Living with partner | 2619 (8.23) | 2236 (8.59) | 383 (6.33) | |
| Married | 16524 (55.24) | 13490 (55.84) | 3034 (51.99) | |
| Never married | 5861 (18.08) | 5033 (19.26) | 828 (11.70) | |
| Separated | 1076 (2.36) | 771 (2.16) | 305 (3.42) | |
| Widowed | 2527 (5.67) | 1533 (4.33) | 994 (12.85) | |
| BMI category no. (Weighted%) | | | | < 0.0001 |
| <25 | 9196 (29.76) | 7848 (31.39) | 1348 (21.03) | |
| 25-30 | 10656 (32.84) | 8754 (33.39) | 1902 (29.90) | |
| >=30 | 12314 (37.40) | 9077 (35.23) | 3237 (49.06) | |
| Smoking status no. (Weighted%) | | | | < 0.0001 |
| Never | 17630 (54.84) | 14469 (56.05) | 3161 (48.36) | |
| Former | 7818 (24.85) | 5873 (23.87) | 1945 (30.11) | |
| Now | 6718 (20.31) | 5337 (20.09) | 1381 (21.53) | |
| Drinking status no. (Weighted% | | | | < 0.0001 |
| Never | 4577 (10.86) | 3447 (10.21) | 1130 (14.36) | |
| Former | 5292 (13.36) | 3720 (11.82) | 1572 (21.62) | |
| Now | 22297 (75.78) | 18512 (77.97) | 3785 (64.02) | |
| Hypertension no. (Weighted%) | | | | < 0.0001 |
| No | 18400 (62.06) | 16088 (66.06) | 2312 (40.60) | |
| Yes | 13766 (37.94) | 9591 (33.94) | 4175 (59.40) | |
| Diabetes no. (Weighted%) | | | | < 0.0001 |
| No | 23452 (77.72) | 19747 (80.61) | 3705 (62.17) | |
| IGT | 1238 (3.52) | 951 (3.41) | 287 (4.14) | |
| IFG | 1468 (4.78) | 1137 (4.55) | 331 (6.01) | |

*(Continued)*

**Table 1.** (Continued)

| Characteristics | Total (n = 32,166) [a] | Individuals without overactive bladder (n = 25,679) [a] | Individuals with overactive bladder (n = 6,487) [a] | P value |
|---|---|---|---|---|
| DM | 6008 (13.98) | 3844 (11.43) | 2164 (27.68) | |

BMI, body mass index; DM, diabetes mellitus; IFG, impaired fasting glycaemia; IGT, impaired glucose tolerance; NHANES, the National Health and Nutrition Examination Survey; SE, standard error.

Mean ± SE for continuous variables, the *P* value was calculated by weighted t test; Number (Weighted%) for categorical variables, the *P* value was calculated by weighted chi-square test. [a]Unweighted number of observations in dataset.

## Relationship between cancer status and OAB

Table 2 reveals the outcomes of logistic regression analyses, elucidating a positive correlation between presence of cancer and the incidence of OAB. Across Model 1 (OR = 2.34; 95% CI: 2.11-2.59; *P* < 0.0001), Model 2 (OR = 1.49; 95% CI: 1.32-1.69; *P* < 0.0001), and Model 3 (OR = 1.45; 95% CI: 1.29-1.64; *P* < 0.0001), individuals with cancer consistently demonstrated an elevated risk of OAB compared to their counterparts without cancer. A comprehensive multivariate regression analysis further identified significant associations of sex, age, race, education level, marital status, BMI, smoking status and drinking status with OAB. Moreover, hypertension (OR = 1.35; 95% CI: 1.23-1.49; *P* < 0.0001) and diabetes (OR = 1.51; 95% CI: 1.36-1.67; *P* < 0.0001) emerged as important factors significantly linked to an increased risk of OAB.

GU cancer specifically encompasses malignancies of the kidney, prostate, bladder, and testis. In contrast, non-GU cancer comprises all other types of cancer not related to the urinary and reproductive systems. As shown in Table S1, the findings indicated that individuals with both GU and non-GU cancer were at an increased risk of developing OAB compared to participants without cancer. Pelvic cancer specifically refers to malignancies affecting the prostate, bladder, colon, cervix (cervical), uterus (uterine), rectum (rectal), and ovaries (ovarian). The non-pelvic cancer category encompasses all other forms of cancer that do not originate in the pelvic region. The findings also indicated that, when compared to individuals without cancer, participants with either pelvic or non-pelvic cancer faced an elevated risk of developing OAB (Table S2).

## Subgroup analysis

Study participants were systematically categorized based on sex, age, BMI, smoking behavior, alcohol consumption patterns, hypertension prevalence, and diabetes status (Fig 2). Results from subgroup analyses revealed affirmative and consistent correlations between cancer and OAB among different subgroups. However, statistically significant interactions were observed between sex, age, hypertension, diabetes, and the prevalence of OAB in our study (all *P* for interaction < 0.001). Further epidemiological and mechanistic studies are needed to more comprehensively explain these results and elucidate the underlying mechanisms.

## Association between OAB and mortality among individuals afflicted with cancer

Within the cohort comprising 3,090 participants diagnosed with cancer, a total of 850 occurrences of all-cause mortality were documented over a median follow-up period spanning 76 months. The primary objective of the study was to assess the impact of OAB on the overall risk of mortality among individuals diagnosed with cancer. Analysis of Kaplan-Meier survival

**Table 2. Relationship between cancer and overactive bladder among participants.**

| Variable | Model 1 | | Model 2 | | Model 3 | |
|---|---|---|---|---|---|---|
| | OR (95% CI) | *P* value | OR (95% CI) | *P* value | OR (95% CI) | *P* value |
| Cancer | | | | | | |
| No | ref | ref | ref | ref | ref | ref |
| Yes | 2.34 (2.11, 2.59) | <0.0001 | 1.49 (1.32, 1.69) | <0.0001 | 1.45 (1.29, 1.64) | <0.0001 |
| Sex | | | | | | |
| Female | | | ref | ref | ref | ref |
| Male | | | 0.62 (0.57, 0.68) | <0.0001 | 0.59 (0.54, 0.65) | <0.0001 |
| Age group | | | | | | |
| ≤49 | | | ref | ref | ref | ref |
| 50-65 | | | 2.77 (2.48, 3.09) | <0.0001 | 2.34 (2.08, 2.63) | <0.0001 |
| ≥65 | | | 5.37 (4.76, 6.06) | <0.0001 | 4.22 (3.69, 4.83) | <0.0001 |
| Race | | | | | | |
| Hispanic | | | ref | ref | ref | ref |
| Non-Hispanic White | | | 0.85 (0.73, 0.98) | 0.03 | 0.82 (0.71, 0.95) | 0.01 |
| Non-Hispanic Black | | | 1.68 (1.45, 1.95) | <0.0001 | 1.53 (1.32, 1.78) | <0.0001 |
| Mexican American | | | 0.85 (0.73, 1.00) | 0.04 | 0.87 (0.74, 1.01) | 0.07 |
| Other | | | 0.93 (0.77, 1.13) | 0.46 | 0.85 (0.69, 1.04) | 0.11 |
| Education | | | | | | |
| Less than high school | | | ref | ref | ref | ref |
| High school or equivalent | | | 0.62 (0.55, 0.69) | <0.0001 | 0.67 (0.60, 0.74) | <0.0001 |
| Some college or AA degree | | | 0.56 (0.49, 0.63) | <0.0001 | 0.62 (0.55, 0.70) | <0.0001 |
| College graduate or above | | | 0.38 (0.33, 0.44) | <0.0001 | 0.47 (0.41, 0.54) | <0.0001 |
| Marital status | | | | | | |
| Divorced | | | ref | ref | ref | ref |
| Living with partner | | | 0.95 (0.80, 1.13) | 0.58 | 0.96 (0.80, 1.14) | 0.61 |
| Married | | | 0.82 (0.73, 0.92) | <0.001 | 0.85 (0.75, 0.95) | 0.005 |
| Never married | | | 0.86 (0.75, 1.00) | 0.05 | 0.91 (0.78, 1.05) | 0.19 |
| Separated | | | 1.30 (1.06, 1.60) | 0.01 | 1.28 (1.04, 1.57) | 0.02 |
| Widowed | | | 0.94 (0.79, 1.12) | 0.48 | 0.92 (0.77, 1.11) | 0.38 |
| BMI category | | | | | | |
| <25 | | | ref | ref | ref | ref |
| 25-30 | | | 1.25 (1.13, 1.38) | <0.0001 | 1.20 (1.08, 1.34) | <0.001 |
| ≥30 | | | 1.91 (1.73, 2.10) | <0.0001 | 1.66 (1.50, 1.84) | <0.0001 |
| Smoking status | | | | | | |
| Never | | | | | ref | ref |
| Former | | | | | 1.14 (0.99, 1.31) | 0.07 |
| Now | | | | | 1.44 (1.28, 1.63) | <0.0001 |
| Drinking status | | | | | | |
| Never | | | | | ref | ref |
| Former | | | | | 1.12 (0.96, 1.30) | 0.16 |
| Now | | | | | 0.86 (0.74, 0.99) | 0.03 |
| Hypertension | | | | | | |
| No | | | | | ref | ref |
| Yes | | | | | 1.35 (1.23, 1.49) | <0.0001 |
| Diabetes | | | | | | |
| No | | | | | ref | ref |
| IGT | | | | | 1.06 (0.88, 1.27) | 0.54 |

*(Continued)*

**Table 2.** (Continued)

| Variable | Model 1 | | Model 2 | | Model 3 | |
|---|---|---|---|---|---|---|
| | OR (95% CI) | *P* value | OR (95% CI) | *P* value | OR (95% CI) | *P* value |
| IFG | | | | | 1.28 (1.02, 1.61) | 0.03 |
| DM | | | | | 1.51 (1.36, 1.67) | <0.0001 |

BMI, body mass index; DM, diabetes mellitus; IFG, impaired fasting glycaemia; IGT, impaired glucose tolerance; NHANES, the National Health and Nutrition Examination Survey.

Model 1, unadjusted crude model.

Model 2, adjusted for demographic characteristics (sex, age group, race, education, marital status); BMI category;

Model 3, adjusted for demographic characteristics (sex, age group, race, education, marital status); BMI category, smoking status, drinking status, hypertension and diabetes.

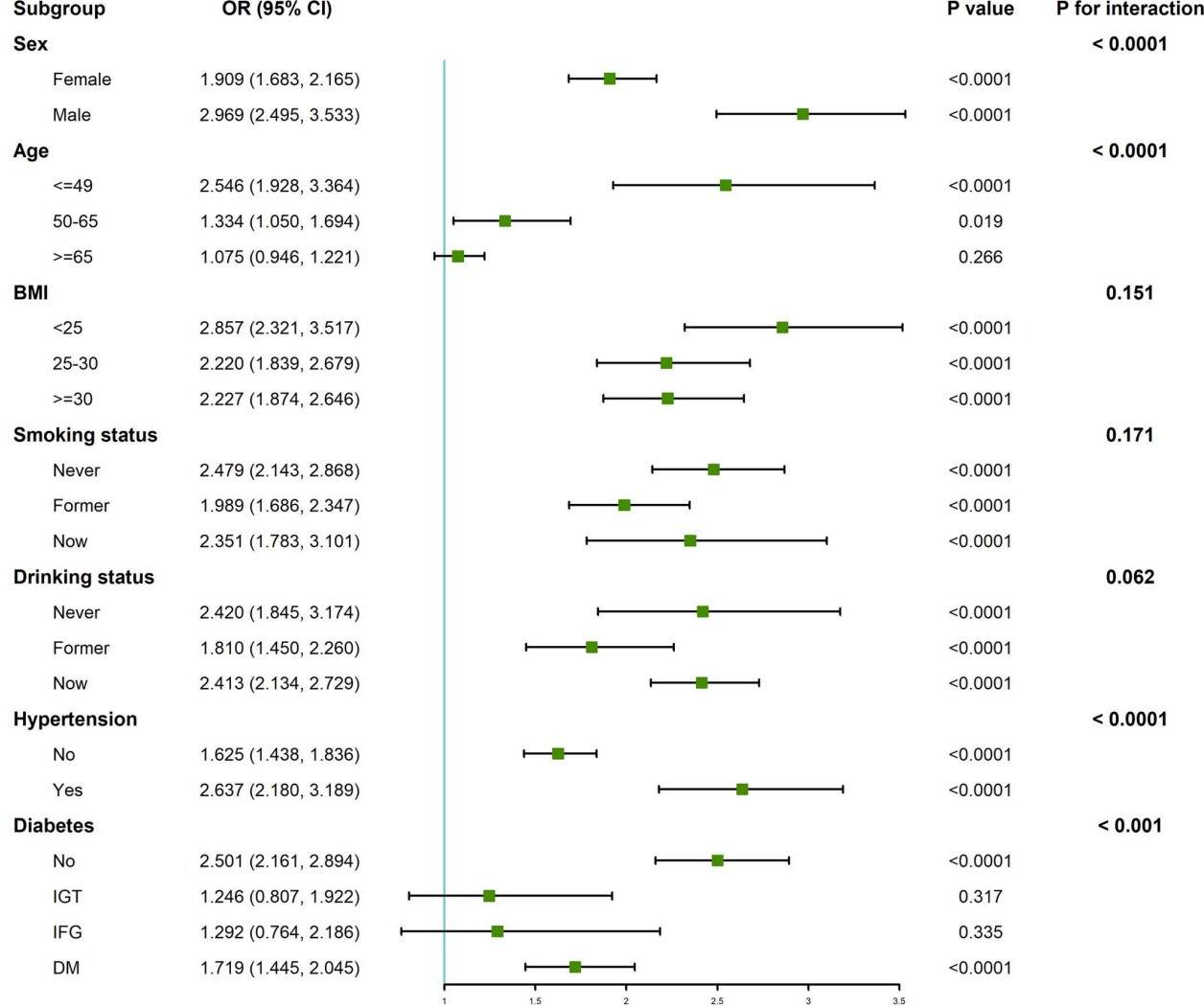

**Fig 2. Forest plot for subgroup analysis between cancer and OAB.**

curves (Fig 3) revealed a statistically significant reduction in survival probability among participants with OAB, as determined by the log-rank test (*P* < 0.0001).

As illustrated in Table 3, the presence of OAB was found to be significantly associated with heightened all-cause mortality risk among individuals diagnosed with cancer, as evidenced by the results of Model 1 (HR: 2.20; 95% CI: 1.83-2.64; P < 0.0001). Upon adjustment for sex, age, race, education level, marital status, and BMI in Model 2, OAB was still linked to increased risk of all-cause mortality in participants diagnosed with cancer (HR: 1.72; 95% CI: 1.45-2.03; *P* < 0.0001). This association persisted in Model 3 (HR: 1.68; 95% CI: 1.42-1.98; *P* < 0.0001), where additional adjustments were made for smoking behavior, alcohol consumption patterns, hypertension prevalence, and diabetes status.

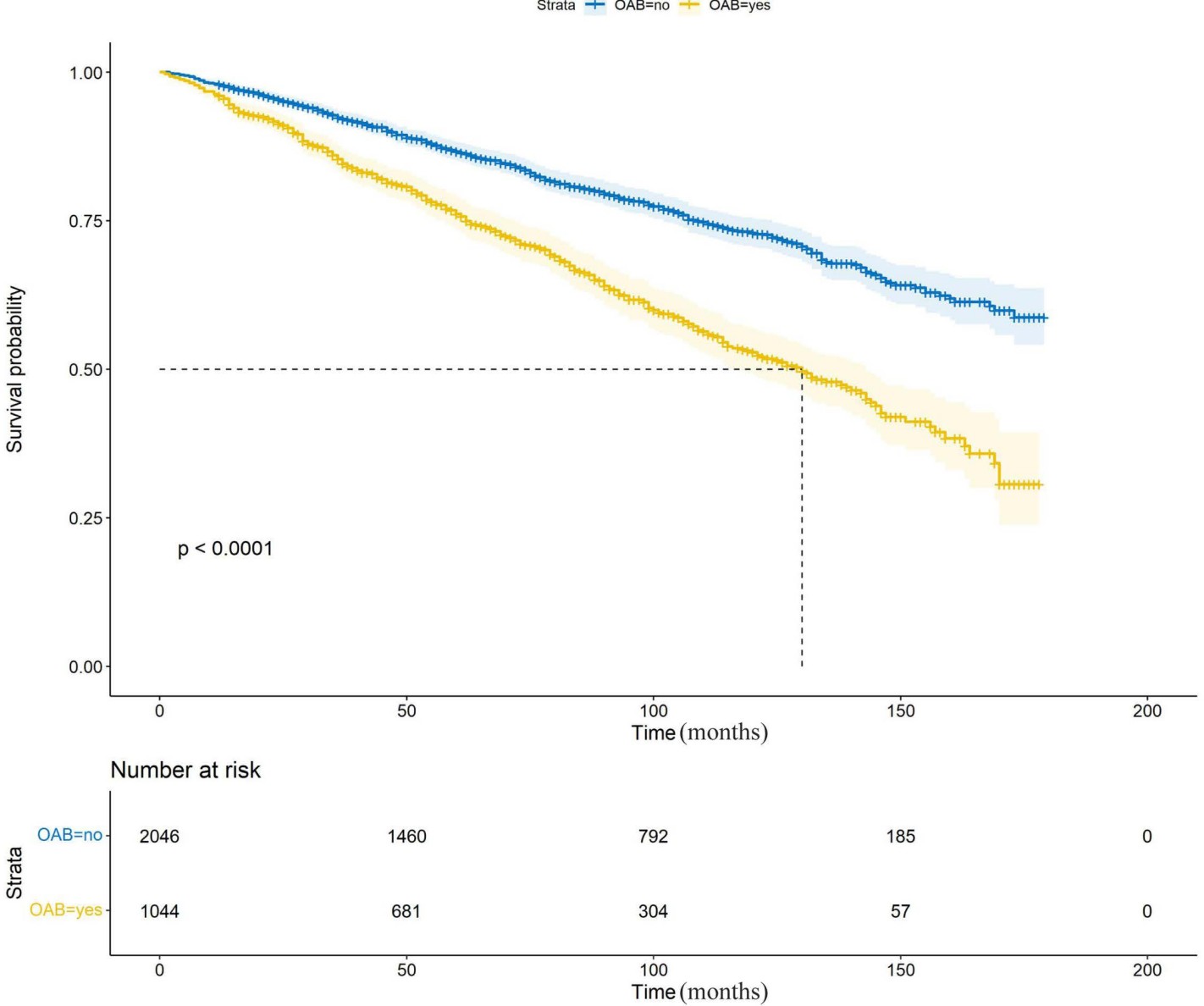

**Fig 3. Association between OAB and all-cause mortality among individuals afflicted with cancer.**

**Table 3. Association of overactive bladder with all-cause death in individuals with cancer among participants.**

| Variable | Model 1 | | Model 2 | | Model 3 | |
|---|---|---|---|---|---|---|
| | HR (95% CI) | *P* value | HR (95% CI) | *P* value | HR (95% CI) | *P* value |
| Overactive bladder | | | | | | |
| No | ref | ref | ref | ref | ref | ref |
| Yes | 2.20 (1.83, 2.64) | <0.0001 | 1.72 (1.45, 2.03) | <0.0001 | 1.68 (1.42, 1.98) | <0.0001 |
| Sex | | | | | | |
| Female | | | ref | ref | ref | ref |
| Male | | | 1.74 (1.44, 2.09) | <0.0001 | 1.73 (1.43, 2.09) | <0.0001 |
| Age group | | | | | | |
| ≤49 | | | ref | ref | ref | ref |
| 50-65 | | | 3.36 (1.75, 6.47) | <0.001 | 2.88 (1.46, 5.66) | 0.002 |
| ≥65 | | | 10.21 (5.82,17.91) | <0.0001 | 8.22 (4.39,15.37) | <0.0001 |
| Race | | | | | | |
| Hispanic | | | ref | ref | ref | ref |
| Non-Hispanic White | | | 2.61 (1.54, 4.44) | <0.001 | 2.68 (1.53, 4.70) | <0.001 |
| Non-Hispanic Black | | | 2.66 (1.55, 4.55) | <0.001 | 2.37 (1.33, 4.25) | 0.004 |
| Mexican American | | | 1.60 (0.90, 2.85) | 0.11 | 1.80 (0.98, 3.31) | 0.06 |
| Other | | | 2.61 (1.30, 5.27) | 0.01 | 2.56 (1.26, 5.21) | 0.01 |
| Education | | | | | | |
| Less than high school | | | ref | ref | ref | ref |
| High school or equivalent | | | 0.73 (0.55, 0.97) | 0.03 | 0.81 (0.62, 1.04) | 0.1 |
| Some college or AA degree | | | 0.55 (0.41, 0.73) | <0.0001 | 0.67 (0.51, 0.89) | 0.005 |
| College graduate or above | | | 0.39 (0.31, 0.51) | <0.0001 | 0.56 (0.44, 0.71) | <0.0001 |
| Marital status | | | | | | |
| Divorced | | | ref | ref | ref | ref |
| Living with partner | | | 0.48 (0.23, 0.97) | 0.04 | 0.46 (0.24, 0.90) | 0.02 |
| Married | | | 0.62 (0.49, 0.79) | <0.0001 | 0.59 (0.46, 0.75) | <0.0001 |
| Never married | | | 0.95 (0.61, 1.47) | 0.82 | 0.88 (0.57, 1.37) | 0.58 |
| Separated | | | 0.84 (0.47, 1.50) | 0.56 | 0.73 (0.40, 1.34) | 0.31 |
| Widowed | | | 1.28 (0.94, 1.75) | 0.12 | 1.21 (0.90, 1.63) | 0.2 |
| BMI category | | | | | | |
| <25 | | | ref | ref | ref | ref |
| 25-30 | | | 0.75 (0.61, 0.93) | 0.01 | 0.71 (0.57, 0.88) | 0.002 |
| ≥30 | | | 0.73 (0.58, 0.92) | 0.01 | 0.62 (0.49, 0.79) | <0.001 |
| Smoking status | | | | | | |
| Never | | | | | ref | ref |
| Former | | | | | 1.37 (1.12, 1.68) | 0.003 |
| Now | | | | | 1.84 (1.38, 2.45) | <0.0001 |
| Drinking status | | | | | | |
| Never | | | | | ref | ref |
| Former | | | | | 1.05 (0.81, 1.36) | 0.71 |
| Now | | | | | 0.63 (0.47, 0.83) | 0.001 |
| Hypertension | | | | | | |
| No | | | | | ref | ref |
| Yes | | | | | 1.59 (1.30, 1.95) | <0.0001 |
| Diabetes | | | | | | |
| No | | | | | ref | ref |
| IGT | | | | | 1.25 (0.87, 1.79) | 0.23 |

*(Continued)*

**Table 3.** (Continued)

| Variable | Model 1 | | Model 2 | | Model 3 | |
|---|---|---|---|---|---|---|
| | HR (95% CI) | *P* value | HR (95% CI) | *P* value | HR (95% CI) | *P* value |
| IFG | | | | | 0.93 (0.58, 1.47) | 0.75 |
| DM | | | | | 1.36 (1.10, 1.69) | 0.01 |

BMI, body mass index; DM, diabetes mellitus; IFG, impaired fasting glycaemia; IGT, impaired glucose tolerance; NHANES, the National Health and Nutrition Examination Survey.

Model 1, unadjusted crude model.

Model 2, adjusted for demographic characteristics (sex, age group, race, education, marital status); BMI category;

Model 3, adjusted for demographic characteristics (sex, age group, race, education, marital status); BMI category, smoking status, drinking status, hypertension and diabetes.

Notably, participants with OAB who did not have cancer exhibited a lower risk of all-cause mortality (HR: 1.42; 95% CI: 1.25-1.60; *P* < 0.0001) (Table S3). Moreover, the influence of OAB on mortality from all causes was observed across various cancer types, including both genitourinary and non-genitourinary, as well as pelvic and non-pelvic cancers (Table S4-7). After accounting for multiple covariates, OAB was also linked to an elevated risk of cancer-specific, cardiovascular disease-related and cerebrovascular disease-related mortality (Table S8-11).

## Discussion

In the current investigation, our objective was to unravel the intricate connections among cancer, OAB, and mortality in adult participants of the United States. This study stands out as one of the largest endeavors to explore the potential relationship between cancer and OAB in adult participants, utilizing NHANES data from 1999 to 2018. Our exhaustive examination unveiled a notable correlation between the presence of cancer and an elevated susceptibility to OAB. Cancer patients are more vulnerable and having more risk factors to get OAB complaints. Furthermore, survival analysis demonstrated a potential association between OAB presence and heightened mortality risk among adult cancer survivors. These findings underscore the need for further in-depth exploration through both basic and clinical research.

As a prevailing chronic ailment on a global scale, OAB not only impacts the physical health and quality of life of afflicted individuals but also gives rise to substantial economic implications characterized by diminished productivity and elevated healthcare expenditures. In the context of the US population, healthcare costs associated with OAB patients were found to exceed those of comparable individuals without OAB by more than 2.5 times [13]. Contemporary literature posits that the nonspecific symptomatology inherent in OAB may be attributed to diverse pathophysiological factors and mechanisms [14]. Prior research has indicated that marijuana exhibits a relieving effect on lower urinary tract symptoms. Nevertheless, it is noteworthy that consistent marijuana usage might concurrently elevate the risk of developing OAB [8]. Lin et al. conducted an extensive investigation aimed at examining the relationship between urinary metabolites of organophosphate esters and OAB. The findings from this study unveil a significant association, indicating that exposure to diphenyl phosphate is correlated with heightened susceptibility to OAB among adult females in the United States [15]. Notably, individuals exhibiting circadian syndrome demonstrated a markedly higher prevalence of OAB when contrasted with those without circadian syndrome [9]. Unraveling the mechanisms that underlie the manifestation of OAB symptoms offers a prospective avenue for tailoring treatments to the specific needs of individual patients, thereby fostering improved clinical outcomes [14].

In the broader context of mortality patterns within the United States, cancer emerges as the second most prominent cause of death, underscored by its significant impact on public health outcomes [3]. With the modern treatment widely used in the clinical practice, the lifespan of cancer survivors has dramatically increased. Thus, we should pay more attention to the quality life of cancer survivors. Noteworthy, there are few studies to evaluate the relationship between cancer and OAB. Thus, our study provides affirmative evidence for the potential relationship between cancer and prevalent OAB.

There are several potential explanations for the association between cancer and OAB. Firstly, it is plausible that some cancer survivors may present with OAB symptoms as clinical manifestations of their prior cancer diagnosis. Secondly, cancer survivors often undergo various treatment modalities, such as surgery, chemotherapy, immunotherapy, and radiotherapy, all of which may contribute to an elevated risk of developing lower urinary tract symptoms and OAB. Thirdly, psychological factors, such as anxiety, nervousness, or other psychological comorbidities commonly experienced by cancer survivors, could also predispose individuals to an increased risk of OAB. Our investigation extends the groundwork laid by preceding longitudinal studies that have delved into the association between cancer and OAB.

When we restrict our analysis to participants with cancer, we observed that OAB was an independent risk of mortality. OAB is more common in participants with older age. A commonly acknowledged rationale posits that the etiological factors contributing to OAB in cancer survivors, including demographic variables such as sex and age, as well as comorbid conditions like hypertension or diabetes, are intricately intertwined with the broader spectrum of mortality risk factors. Furthermore, it is plausible that OAB among the elderly population could be correlated with heightened susceptibility to falls and fractures, factors that potentially exert an impact on the overall risk of mortality across various causes [16]. In the clinical practice, anticholinergic medications are widely used to treat patients with OAB. Numerous extensive observational inquiries have been conducted to ascertain the potential association between anticholinergic medications and elevated mortality risks [16]. Notable findings from investigations conducted among diverse cohorts of elderly individuals underscore a significant observation: the use of oxybutynin demonstrates a substantially elevated risk of mortality, ranging from 26% to 58%, compared to alternative anticholinergic medications or β3 agonists prescribed for the management of OAB. Thus, for cancer survivors with OAB, we should pay attention to the side effects of anticholinergic medications, evaluate the healthy condition of cancer survivors and choose more safe drugs for them.

As the most comprehensive investigation conducted to date examining the correlation between cancer and OAB, this study boasts several notable methodological strengths. Firstly, our analysis leveraged nationally representative survey data from the NHANES. The substantial sample size inherent in NHANES facilitated robust subgroup analyses and allowed for meticulous adjustment for potential confounding variables. Secondly, the participants in our study were subject to random sampling from the general population, ensuring the representativeness and generalizability of our findings. Consequently, our results can be confidently extrapolated to the general population of the US. Thirdly, the extensive sample size, high event rates, and prolonged follow-up duration of our investigation enabled a thorough examination of the correlation between OAB and mortality among individuals with cancer.

It is essential to contextualize our results within the confines of particular limitations. Firstly, it's important to acknowledge that the cross-sectional nature of the NHANES data presents a methodological constraint, precluding the unequivocal determination of causality between cancer and OAB. Secondly, despite our efforts to adjust for relevant confounding variables, there may still be residual confounding from unmeasured or unknown factors. For example, these participants might have multiple diseases, such as chronic obstructive pulmonary disease, chronic liver disease, and chronic kidney disease, which might also affect mortality. However, we

accounted for potential survival predictors to mitigate the influence of confounding variables. Despite these constraints, our investigation unveiled a positive association between cancer, OAB and mortality. Future research with larger sample sizes and more precise measurement methods is crucial to establishing a more definitive causal connection between cancer and OAB.

## Conclusions

The investigation identified a noteworthy positive correlation between cancer and the likelihood of OAB. Moreover, OAB was significantly correlated with an elevated risk of mortality among individuals who had received a cancer diagnosis. Effective treatments for OAB may represent viable strategies for lowering mortality risk among patients with cancer, bearing potential implications for both clinical practice and public health.

## Supporting Information

**Table S1. Relationship between Non-GU and GU cancer and overactive bladder among participants.**
(DOCX)

**Table S2. Relationship between non-pelvic and pelvic cancer and overactive bladder among participants.**
(DOCX)

**Table S3. Association of overactive bladder with all-cause death among participants without cancer.**
(DOCX)

**Table S4. Association of overactive bladder with all-cause mortality among participants with GU cancer.**
(DOCX)

**Table S5. Association of overactive bladder with all-cause mortality among participants with Non-GU cancer.**
(DOCX)

**Table S6. Association of overactive bladder with all-cause mortality among participants with pelvic cancer.**
(DOCX)

**Table S7. Association of overactive bladder with all-cause mortality among participants with non-pelvic cancer.**
(DOCX)

**Table S8. Association of overactive bladder with cancer-specific mortality among participants with cancer.**
(DOCX)

**Table S9. Association of overactive bladder with cardiovascular disease-related mortality among participants with cancer.**
(DOCX)

**Table S10. Association of overactive bladder with cerebrovascular disease-related mortality among participants with cancer.**
(DOCX)

**Table S11. Causes of death among participants with cancer.**
(DOCX)

AcknowledgmentThanks to Zhang Jing (Second Department of Infectious Disease, Shanghai Fifth People's Hospital, Fudan University) for his work on the NHANES database. His outstanding work, the nhanesR package and webpage, makes it easier for us to explore the NHANES database.

## Author contributions

**Conceptualization:** Zhaowei Zhu, Xuepei Zhang.

**Data curation:** Bin Zhi, Pin Zhao.

**Formal analysis:** Bin Zhi.

**Methodology:** Liyuan Duan.

**Resources:** Bin Zhi, Liyuan Duan.

**Software:** Bin Zhi, Pin Zhao, Liyuan Duan.

**Validation:** Pin Zhao, Yuan Liu.

**Visualization:** Pin Zhao, Yuan Liu.

**Writing – original draft:** Bin Zhi, Pin Zhao.

**Writing – review & editing:** Zhaowei Zhu, Xuepei Zhang.

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
