## [Decision Letter · Decision Letter 0]

15 Oct 2024

PONE-D-24-20411Association of cancer with overactive bladder and impact of overactive bladder on mortality among cancer survivors: NHANES 1998-2018PLOS ONE

Dear Dr. Zhu,

Thank you for submitting your manuscript to PLOS ONE. After careful consideration, we feel that it has merit but does not fully meet PLOS ONE’s publication criteria as it currently stands. Therefore, we invite you to submit a revised version of the manuscript that addresses the points raised during the review process.

We look forward to receiving your revised manuscript.

Kind regards,

Li Yang, M.D.

Academic Editor

PLOS ONE

Journal Requirements:

**Additional Editor Comments:**

Thanks for submitting your work to PLOS ONE. Your manuscript has now been assessed by our editorial team and external peer experts. You can see that they have proposed many serious problems and are advising you should revise your paper comprehensively. Thus, we invite you to undertake the required revision work. If these issues are properly addressed by authors and approved by reviewers, we would be pleased to reconsider the decision for publication of your work. Please note that this revision decision does not assure the final acceptance. Thanks for the chance to consider your work. 

Reviewers' comments:

Reviewer's Responses to Questions

**Comments to the Author**

1. Is the manuscript technically sound, and do the data support the conclusions?

Reviewer #1: Partly

Reviewer #2: Partly

2. Has the statistical analysis been performed appropriately and rigorously? 

Reviewer #1: I Don't Know

Reviewer #2: Yes

3. Have the authors made all data underlying the findings in their manuscript fully available?

Reviewer #1: Yes

Reviewer #2: No

4. Is the manuscript presented in an intelligible fashion and written in standard English?

Reviewer #1: Yes

Reviewer #2: Yes

5. Review Comments to the Author

Reviewer #1: 1- I congratulate the authors for the study conducted on a large sample. However, I have some suggestions regarding the article.

2- The relationship between mortality due to cerebrovascular and cardiovascular diseases is given in Tables S8-10, but data on patient numbers can be added to Table 1.

3- The study analyzed the relationship between mortality due to cerebrovascular and cardiovascular diseases. Were there any patients with diseases that could affect mortality, such as chronic obstructive pulmonary disease, chronic liver disease, and chronic kidney disease, in the patient group included in the study? If so, why were they not included in the analysis?

4- What were the treatments for overactive bladder? Were there any patients who did not receive OAB treatment?

5- The relationship between mortality and side effects of the chemotherapeutic agents used can be analyzed (e.g. kidney damage due to cisplatin use).

6 - In the discussion section, additional information about the lower urinary tract symptoms that may occur due to radiotherapy and some chemotherapeutics in cancer patients can be given.

Reviewer #2: The authors determine the correlation between cancer disease and a chronic syndrome of over active bladder. They quesion "the impact of OAB on mortality". I appreciate that the authors try to increase the awareness for OAB in this patiënt group. I have some questions and comments.

Introduction:

- "there is a pressing need ... to improve their survival...". Is it really only to improve survival or to get awareness of OAB in these patients to adequately treat these bothersome complaints and increase qualitity of life in cancer disease/survivors?

- Is het hypothesis of this study that OAB (as risk factors) increase mortality?

Methods:

- NHANES cycles spanning from 1999-2000 to 2017-2018 => mention exact start/closure.

- Remove "variable meq220), espescially as the database have not been made available in the submission or been mention in what way availlable correctly.

- Outcome variable: validated questions on its own?

- OABSS, "a rigorously"=> validated? Cumulative OABSS of UUI and nocturia score=> validated outcome?

- Covariate assessment/statistical analysis: 4x "meticulously". Avoid subjective words in methods/results anyway.

Results:

- "9.606 and 20.167%=> 10 and 20% or 9.6 and 20.2%. Idem for mean age/BMI etcetera.

- Individuals with OAB had more know risk factors for OAB: age, BMI, diabetes mellitus, alcohol consumption, hypertension. => implies that the cancer disease population is a more vulnerable population with more morbidity and risk factors for OAB? And OAB symptoms to be aware of as it is already suspected to occur more in a patient?

- Relationship between cancer status and OAB: Across model: not explanted in text methods and not in al tables.

- "we categorized cancer patients..." and "Furthermore, the study participants..."=> methods section.

Discussion:

- "demonstrated a robust association between OAB presence and heightened mortality": OAB not an entity/disease itself. No causality. Correlation the other way around? Cancer patients more vulnerable and having more risk factors to get OAB complaints?

Tables/figures:

- Figure 3: units at axes

6. PLOS authors have the option to publish the peer review history of their article (what does this mean?). If published, this will include your full peer review and any attached files.

Reviewer #1: No

Reviewer #2: **Yes: **Frank Martens

---

## [Author Response · Author response to Decision Letter 0]

29 Nov 2024

Reviewer #1: 

Comment 1: I congratulate the authors for the study conducted on a large sample. However, I have some suggestions regarding the article.

Answer 1: We greatly appreciate the reviewer’s valuable comment. We would like to revise the manuscript carefully according to the reviewer’s insightful suggestion.

Comment 2: The relationship between mortality due to cerebrovascular and cardiovascular diseases is given in Tables S8-10, but data on patient numbers can be added to Table 1.

Answer 2: Thank you very much for the comment. We provide the association of overactive bladder with cancer-specific mortality, cardiovascular disease-related mortality and cerebrovascular disease-related mortality among participants with cancer in Table S8-10. In Table 1, we provide the general characteristics of individuals stratified by the presence or absence of overactive bladder. We greatly appreciate the reviewer’s valuable suggestion, but we think it is not proper to add these patient numbers in Table 1. Instead, we add another table (Table S11) to show the exact patient numbers about the causes of death among participants with cancer.

Comment 3: The study analyzed the relationship between mortality due to cerebrovascular and cardiovascular diseases. Were there any patients with diseases that could affect mortality, such as chronic obstructive pulmonary disease, chronic liver disease, and chronic kidney disease, in the patient group included in the study? If so, why were they not included in the analysis?

Answer 3: Thank you very much for the insightful comment. Actually, these participants might have multiple diseases that could affect mortality. However, it is impossible to have detailed information of each participant due to the intrinsic limitations of NHANES database. Thus, we stated in the Discussion section that there may still be residual confounding from unmeasured or unknown factors. Despite these constraints, our investigation unveiled a positive association between cancer, OAB and mortality. We greatly appreciate the reviewer’s comment and have added these potential factors in the Discussion section.

Comment 4: What were the treatments for overactive bladder? Were there any patients who did not receive OAB treatment?

Answer 4: Thank you very much for the considerate comment. However, the NHANES database does not provide detailed information about the treatments for overactive bladder.

Comment 5: The relationship between mortality and side effects of the chemotherapeutic agents used can be analyzed (e.g. kidney damage due to cisplatin use).

Answer 5: We greatly appreciate the reviewer’s insightful comment. The relationship between mortality and side effects of the chemotherapeutic agents is a meaningful topic and warrants further investigation. However, the NHANES database did not provide such information. Thus, we could not explore such relationship using the NHANES database. We will pay attention to this question and try to investigate the relationship in future studies.

Comment 6: In the discussion section, additional information about the lower urinary tract symptoms that may occur due to radiotherapy and some chemotherapeutics in cancer patients can be given.

Answer 6: Thank you very much for the valuable suggestion. We have added the statement about the lower urinary tract symptoms due to radiotherapy and some chemotherapeutics in the Discussion section.

Reviewer #2: The authors determine the correlation between cancer disease and a chronic syndrome of over active bladder. They question "the impact of OAB on mortality". I appreciate that the authors try to increase the awareness for OAB in this patient group. I have some questions and comments.

Introduction:

Comment 1: "there is a pressing need ... to improve their survival...". Is it really only to improve survival or to get awareness of OAB in these patients to adequately treat these bothersome complaints and increase quality of life in cancer disease/survivors?

Answer 1: Thank you very much for the constructive comment. We have revised the Introduction section according to the reviewer’s valuable suggestion.

Comment 2: Is the hypothesis of this study that OAB (as risk factors) increase mortality?

Answer 2: We totally agree with the reviewer. The initial hypothesis is that OAB might increase mortality in patients with cancer. After accounting for multiple covariates, OAB was associated with an elevated risk of all-cause and cancer-specific mortality among participants diagnosed with cancer.

Methods:

Comment 3: NHANES cycles spanning from 1999-2000 to 2017-2018 => mention exact start/closure.

Answer 3: We greatly appreciate the reviewer’s considerate comment. However, the most recent NHANES began in 1999, and there are two years in one cycle (https://wwwn.cdc.gov/nchs/nhanes/default.aspx). The NHANES database did not mention exact start/closure. Instead, we mention the NHANES cycles which were used in our study.

Comment 4: Remove "variable meq220), especially as the database have not been made available in the submission or been mention in what way available correctly.

Answer 4: We completely agree with the reviewer and have removed “variable mcq220” in the sentence.

Comment 5: Outcome variable: validated questions on its own?

Answer 5: Thank you for the insightful comment. The outcome variable in the present study is OAB. However, the evaluation of OAB involved these specific questions. Actually, these questions were obtained from the NHANES (https://wwwn.cdc.gov/Nchs/Nhanes/2009-2010/KIQ_U_F.htm#KIQ042). We provided the questions in our manuscript. We cherish the hope that other researchers would understand the definition of OAB better.

Comment 6: OABSS, "a rigorously"=> validated? Cumulative OABSS of UUI and nocturia score=> validated outcome?

Answer 6: Thank you very much for the comment. Actually, OABSS is a rigorously developed questionnaire and has been validated by many previous studies. We are sorry for the confusion and have revised this sentence.

Comment 7: Covariate assessment/statistical analysis: 4x "meticulously". Avoid subjective words in methods/results anyway.

Answer 7: We totally agree with the reviewer and have deleted these subjective words in the revised manuscript.

Results:

Comment 8: "9.606 and 20.167%=> 10 and 20% or 9.6 and 20.2%. Idem for mean age/BMI etcetera.

Answer 8: Thank you very much for the constructive comment. We have revised the manuscript according to the reviewer’s valuable suggestion.

Comment 9: Individuals with OAB had more know risk factors for OAB: age, BMI, diabetes mellitus, alcohol consumption, hypertension. => implies that the cancer disease population is a more vulnerable population with more morbidity and risk factors for OAB? And OAB symptoms to be aware of as it is already suspected to occur more in a patient?

Answer 9: We totally agree with the reviewer that the cancer disease population is a more vulnerable population with more morbidity and risk factors for OAB. By analyzing the risk factors of OAB, we could identify patients with OAB earlier.

Comment 10: Relationship between cancer status and OAB: Across model: not explanted in text methods and not in al tables.

Answer 10: Thank you very much for the comment. We have explained the relationship between cancer status and OAB in the “Statistical Analysis” of Methods section. Moreover, the relationship across models was also shown in Table 2 and Results section.

Comment 11: "we categorized cancer patients..." and "Furthermore, the study participants..."=> methods section.

Answer 11: We greatly appreciate the valuable comment and have moved these two sentences to the Methods section.

Discussion:

Comment 12: "demonstrated a robust association between OAB presence and heightened mortality": OAB not an entity/disease itself. No causality. Correlation the other way around? Cancer patients more vulnerable and having more risk factors to get OAB complaints?

Answer 12: Thank you very much for the constructive comment. We totally agree with the reviewer and have revised this paragraph to make it make more plausible.

Tables/figures:

Comment 13: Figure 3: units at axes

Answer13: We greatly appreciate the reviewer’s constructive comment and have added units at axes.

---

## [Decision Letter · Decision Letter 1]

22 Dec 2024

PONE-D-24-20411R1Association of cancer with overactive bladder and impact of overactive bladder on mortality among cancer survivors: NHANES 1999-2018PLOS ONE

Dear Dr. Zhu,

Thank you for submitting your manuscript to PLOS ONE. After careful consideration, we feel that it has merit but does not fully meet PLOS ONE’s publication criteria as it currently stands. Therefore, we invite you to submit a revised version of the manuscript that addresses the points raised during the review process.

We look forward to receiving your revised manuscript.

Kind regards,

Li Yang, M.D.

Academic Editor

PLOS ONE

Additional Editor Comments:

Please further address the concerns of the peer experts.

Reviewers' comments:

Reviewer's Responses to Questions

**Comments to the Author**

1. If the authors have adequately addressed your comments raised in a previous round of review and you feel that this manuscript is now acceptable for publication, you may indicate that here to bypass the “Comments to the Author” section, enter your conflict of interest statement in the “Confidential to Editor” section, and submit your "Accept" recommendation.

Reviewer #1: (No Response)

Reviewer #3: (No Response)

Reviewer #4: (No Response)

2. Is the manuscript technically sound, and do the data support the conclusions?

Reviewer #1: (No Response)

Reviewer #3: (No Response)

Reviewer #4: Yes

3. Has the statistical analysis been performed appropriately and rigorously? 

Reviewer #1: (No Response)

Reviewer #3: (No Response)

Reviewer #4: No

4. Have the authors made all data underlying the findings in their manuscript fully available?

Reviewer #1: (No Response)

Reviewer #3: (No Response)

Reviewer #4: Yes

5. Is the manuscript presented in an intelligible fashion and written in standard English?

Reviewer #1: (No Response)

Reviewer #3: (No Response)

Reviewer #4: No

6. Review Comments to the Author

Reviewer #1: (No Response)

Reviewer #3: 1. In the NHANES database, we often take PIR as an important covariate, which is recommended to be included;

2. BMI as an important risk factor, the team has considered the impact of BMI as a categorical variable on the outcome, but is the impact of BMI as a continuous variable still highly consistent?

3. does OAB value obey normal distribution and what test method is used?

4. is there any missing value in the data included in this article? If yes, what method is used to deal with missing values and what is the basis?

5. the relationship between risk factors and outcomes is significantly positive, but is this relationship linear or nonlinear?

6. the blood pressure value of hypertensive people is: systolic blood pressure greater than or equal to 140mmHg and diastolic blood pressure greater than 90mmHg. Is this standard appropriate?

7. is there any basis for selecting 50 and 65 years old as nodes in Table 2?

8. in Figure 2, the interaction test of gender, age, hypertension and diabetes is less than 0.05. Can existing theories or potential mechanisms explain this result

Reviewer #4: This paper aimed to explore the association of cancer with overactive bladder and impact of overactive bladder on mortality among cancer survivors by using tne data from NHANES 1999-2018. This work is interesting, but some issues came to my attention.

1. Complex Survey Design: As highlighted on the CDC/NHANES website, "For NHANES datasets, the use of sampling weights and sample design variables is necessary to obtain unbiased estimates and accurate standard errors and confidence intervals." The authors must use the complex survey design parameters—specifically strata, clusters, and weights—to ensure valid and reliable results.

2. Missing Data Analysis: The authors must address the issue of missing data by conducting a thorough missing data analysis. Tables should be included to display the percentage of missing values and the patterns of missingness, especially given that the percentage of missing data may likely exceed 40%. Addressing this is essential for evaluating the robustness of the results.

3. There are numerous indicators in the NHANES database, which can be reached a relationship between each other. How did you accout for the coincidence possibility of your work?

4. NHANES database is a retrospective pooled tool, which is mainly obtained from the outcomes of questionnaire survey. Thus, the data may be not reliable enough to be analyzed. Some limitations regarding this issue must be specified.

5. Please consider to add a table to summarize the ongong or completed clinical trials regarding this topic, which may further rich the overall contents of this paper.

6. Please consider to include more cycles of cancer patients in the NHANES database, since current 1999-2018 cycle seemed a little outdated.

7. PLOS authors have the option to publish the peer review history of their article (what does this mean?). If published, this will include your full peer review and any attached files.

Reviewer #1: No

Reviewer #3: No

Reviewer #4: No

---

## [Author Response · Author response to Decision Letter 1]

4 Feb 2025

Reviewer #3:

Comment 1. In the NHANES database, we often take PIR as an important covariate, which is recommended to be included;

Answer 1: Thank you very much for the insightful comment. We appreciate your reference to the Family Income-to-Poverty Ratio (PIR) as an index of socioeconomic standing, which has indeed been utilized in some NHANES studies. However, in the context of our current research, our primary focus is on elucidating the associations between cancer and the risk of Overactive Bladder (OAB), as well as the links between OAB and mortality outcomes among cancer survivors. Drawing on our extensive clinical experience, we believe that PIR may not significantly influence these specific relationships. Instead, we have incorporated a comprehensive set of other important and relevant variables, such as gender, race, education, marital status, BMI, smoking status, drinking status, and history of hypertension and diabetes. We are confident that these combined variables collectively provide a more nuanced and accurate representation of an individual's socioeconomic standing in the context of our study.

Comment 2. BMI as an important risk factor, the team has considered the impact of BMI as a categorical variable on the outcome, but is the impact of BMI as a continuous variable still highly consistent?

Answer 2: Thank you very much for your valuable feedback. In response to your comment, we have conducted a reanalysis of our data to thoroughly assess the impact of BMI as a continuous variable. Our findings indicate that the influence of BMI remains highly consistent when treated as a continuous variable. We recognize that categorizing BMI into distinct groups aligns well with established clinical guidelines and enhances the interpretability of results for both researchers and practitioners. Given these important considerations, we have chosen to maintain BMI as a categorical variable in the present study.

Comment 3. does OAB value obey normal distribution and what test method is used?

Answer 3: We sincerely appreciate the reviewer’s insightful comment. To assess the distribution of OAB values, we employed the Kolmogorov-Smirnov test and found that the overall OAB values do not follow a normal distribution. In our study, OAB is defined such that participants with a cumulative OAB Symptom Score (OABSS) ≥ 3 are categorically diagnosed with OAB. Among the 32,166 participants, 25,679 did not have OAB, while 6,487 were diagnosed with OAB. Notably, among those with OAB, the majority had an OAB score of 3, with fewer patients having scores of 4 or higher. Although the OAB values do not conform to a normal distribution, we believe this pattern is highly representative of the clinical reality. It aligns well with our extensive clinical experience, reflecting the typical distribution of OAB severity that we commonly encounter in practice.

Comment 4. is there any missing value in the data included in this article? If yes, what method is used to deal with missing values and what is the basis?

Answer 4: Thank you very much for your valuable comment. We fully agree with the reviewer that the initial dataset contains missing values due to the inherent limitations of the NHANES database, which could potentially influence subsequent analyses. In preparation for this study, we reviewed numerous articles focusing on NHANES data and observed how other researchers addressed similar challenges. To ensure the robustness and reliability of our findings, we made a decision to include only adults who had complete information on cancer status, OAB, socioeconomic, and health-related characteristics. We recognize that this approach may introduce some selection bias, and we have carefully addressed these limitations in detail in the Discussion section of our manuscript.

Comment 5. the relationship between risk factors and outcomes is significantly positive, but is this relationship linear or nonlinear?

Answer 5: We are very grateful for the reviewer’s considerate comment. We concur with the reviewer that the relationship between risk factors and outcomes is significantly positive. However, based on our study design and the results we have obtained, we have determined that this relationship is generally nonlinear.

Comment 6. the blood pressure value of hypertensive people is: systolic blood pressure greater than or equal to 140mmHg and diastolic blood pressure greater than 90mmHg. Is this standard appropriate?

Answer 6: Thank you very much for your valuable comment. The criteria for defining blood pressure values in hypertensive individuals are consistent with those used in numerous previous studies, treatment guidelines, and clinical practice.

Comment 7. is there any basis for selecting 50 and 65 years old as nodes in Table 2?

Answer 7: Thank you very much for your considerate comment. We fully agree with the reviewer that any choice of age nodes may introduce some bias. However, our decision to select 50 and 65 years as the nodes is based on extensive review of relevant studies. We believe that this categorization aligns more closely with clinical practice and provides a meaningful framework for our analysis.

Comment 8. in Figure 2, the interaction test of gender, age, hypertension and diabetes is less than 0.05. Can existing theories or potential mechanisms explain this result

Answer 8: We greatly appreciate the reviewer’s considerate comment. We fully agree with the reviewer that the interaction tests for gender, age, hypertension, and diabetes all yielded p-values less than 0.05. These findings suggest significant interactions among these variables, which we believe provide important insights and directions for future research. We emphasize that further epidemiological and mechanistic studies are needed to more comprehensively explain these results and elucidate the underlying mechanisms.

Reviewer #4: This paper aimed to explore the association of cancer with overactive bladder and impact of overactive bladder on mortality among cancer survivors by using tne data from NHANES 1999-2018. This work is interesting, but some issues came to my attention.

Comment 1. Complex Survey Design: As highlighted on the CDC/NHANES website, "For NHANES datasets, the use of sampling weights and sample design variables is necessary to obtain unbiased estimates and accurate standard errors and confidence intervals." The authors must use the complex survey design parameters—specifically strata, clusters, and weights—to ensure valid and reliable results.

Answer 1: Thank you very much for the constructive comment. We totally agreed with the reviewer and have employed suitable weighting procedures to derive nationally representative estimates.

Comment 2. Missing Data Analysis: The authors must address the issue of missing data by conducting a thorough missing data analysis. Tables should be included to display the percentage of missing values and the patterns of missingness, especially given that the percentage of missing data may likely exceed 40%.

Answer 2: We greatly appreciate the reviewer’s considerate comment. We fully agree that the initial dataset contains missing values due to the inherent limitations of the NHANES database, and addressing this issue is crucial for evaluating the robustness of our results. To provide transparency, we have clearly illustrated the selection of participants with complete data in Figure 1. Additionally, we have addressed the issue of missing data and provided the percentage of missing values in the revised manuscript. Moreover, we have thoroughly discussed these limitations and their potential impact in the Discussion section of our manuscript.

Comment 3. There are numerous indicators in the NHANES database, which can be reached a relationship between each other. How did you accout for the coincidence possibility of your work?

Answer 3: Thank you very much for the constructive comment. We completely agree with the reviewer that the NHANES database contains a vast array of indicators, many of which may be interrelated. In our clinical practice, we have encountered numerous cancer patients who also exhibit symptoms of OAB. These symptoms may arise from the cancer itself or from cancer-related treatments, including various medications. However, the precise relationship between cancer, OAB, and their impact on patient survival remains unclear. Fortunately, the NHANES database offers a valuable opportunity to explore these questions and seek answers. We believe our study will provide important insights and guidance for the treatment of cancer patients and the improvement of their prognosis.

Comment 4. NHANES database is a retrospective pooled tool, which is mainly obtained from the outcomes of questionnaire survey. Thus, the data may be not reliable enough to be analyzed. Some limitations regarding this issue must be specified.

Answer 4:

Thank you very much for your constructive comment. We fully agree with the reviewer that the NHANES database is a retrospective, pooled tool primarily derived from questionnaire survey outcomes. While it does have some limitations in terms of data analysis, it remains an invaluable resource for addressing important clinical and public health questions. Many researchers have utilized the NHANES database to explore various relationships and provide guidance for further studies. Notably, since 2010, more than 56,000 relevant studies based on NHANES data have been published. This underscores its significance in the research community.

PubMed Search for NHANES Studies (2010-2025):

https://pubmed.ncbi.nlm.nih.gov/?term=National+Health+and+Nutrition+Examination+Survey&filter=years.2010-2025&sort=date

Comment 5. Please consider to add a table to summarize the ongoing or completed clinical trials regarding this topic, which may further rich the overall contents of this paper.

Answer 5: We are extremely grateful for the reviewer’s thoughtful comments. We have conducted an extensive search online but were unable to locate any clinical trials that specifically focus on this topic. We believe this is an emerging area that has not yet garnered widespread attention. We are convinced that our research will provide valuable insights and meaningful directions for future clinical trials in this domain.

Comment 6. Please consider to include more cycles of cancer patients in the NHANES database, since current 1999-2018 cycle seemed a little outdated.

Answer 6: Thank you very much for the valuable suggestion. We appreciate your attention to the timeliness of the data. While it is true that the 1999-2018 cycle of the NHANES database may appear to be somewhat dated, it is essential to recognize that this period of data remains highly relevant and has been extensively utilized in numerous studies. Moreover, our research goes beyond merely examining the relationship between variables; we aim to delve into the impact of OAB on the survival of cancer patients. Including the most recent cycles of NHANES data would result in a follow-up period that is too short to accurately assess the long-term influence of OAB on the prognosis of cancer patients. Therefore, we believe that utilizing the 1999-2018 cycle is more appropriate, as it ensures the accuracy and representativeness of the cohort. Through meticulous research design and robust analytical methods, we have successfully extracted valuable insights from the existing data, providing a solid foundation for future cancer research.

---

## [Decision Letter · Decision Letter 2]

20 Feb 2025

Association of cancer with overactive bladder and impact of overactive bladder on mortality among cancer survivors: NHANES 1999-2018

PONE-D-24-20411R2

Dear Dr. Zhu,

We’re pleased to inform you that your manuscript has been judged scientifically suitable for publication and will be formally accepted for publication once it meets all outstanding technical requirements.

Kind regards,

Li Yang, M.D.

Academic Editor

PLOS ONE

Additional Editor Comments (optional):

Thanks for authors' efforts to respond to my and reviewers' comments. This paper can be accepted now.

Reviewers' comments:

Reviewer's Responses to Questions

**Comments to the Author**

1. If the authors have adequately addressed your comments raised in a previous round of review and you feel that this manuscript is now acceptable for publication, you may indicate that here to bypass the “Comments to the Author” section, enter your conflict of interest statement in the “Confidential to Editor” section, and submit your "Accept" recommendation.

Reviewer #4: All comments have been addressed

2. Is the manuscript technically sound, and do the data support the conclusions?

Reviewer #4: Yes

3. Has the statistical analysis been performed appropriately and rigorously? 

Reviewer #4: Yes

4. Have the authors made all data underlying the findings in their manuscript fully available?

Reviewer #4: Yes

5. Is the manuscript presented in an intelligible fashion and written in standard English?

Reviewer #4: Yes

6. Review Comments to the Author

Reviewer #4: Thanks for your response to my concerns.

7. PLOS authors have the option to publish the peer review history of their article (what does this mean?). If published, this will include your full peer review and any attached files.

Reviewer #4: No

---

## [Editor Report · Acceptance letter]

PONE-D-24-20411R2

PLOS ONE

Dear Dr. Zhu,

I'm pleased to inform you that your manuscript has been deemed suitable for publication in PLOS ONE. Congratulations! Your manuscript is now being handed over to our production team.

Kind regards,

on behalf of

Dr. Li Yang

Academic Editor

PLOS ONE